# Use of Biolayer Interferometry to Identify Dominant Binding Epitopes of Influenza Hemagglutinin Protein of A(H1N1)pdm09 in the Antibody Response to 2010–2011 Influenza Seasonal Vaccine

**DOI:** 10.3390/vaccines11081307

**Published:** 2023-07-31

**Authors:** Zhu Guo, Xiuhua Lu, Paul J. Carney, Jessie Chang, Wen-pin Tzeng, Ian A. York, Min Z. Levine, James Stevens

**Affiliations:** Influenza Division, National Center for Immunization and Respiratory Diseases, Centers for Disease Control and Prevention, 1600 Clifton Road, Atlanta, GA 30329, USA; xal0@cdc.gov (X.L.); gul1@cdc.gov (P.J.C.); vnj9@cdc.gov (J.C.); wdt6@cdc.gov (W.-p.T.); ite1@cdc.gov (I.A.Y.); mwl2@cdc.gov (M.Z.L.)

**Keywords:** influenza vaccine, humoral response, dominant binding epitope

## Abstract

The globular head domain of influenza virus surface protein hemagglutinin (HA1) is the major target of neutralizing antibodies elicited by vaccines. As little as one amino acid substitution in the HA1 can result in an antigenic drift of influenza viruses, indicating the dominance of some epitopes in the binding of HA to polyclonal serum antibodies. Therefore, identifying dominant binding epitopes of HA is critical for selecting seasonal influenza vaccine viruses. In this study, we have developed a biolayer interferometry (BLI)-based assay to determine dominant binding epitopes of the HA1 in antibody response to influenza vaccines using a panel of recombinant HA1 proteins of A(H1N1)pdm09 virus with each carrying a single amino acid substitution. Sera from individuals vaccinated with the 2010–2011 influenza trivalent vaccines were analyzed for their binding to the HA1 panel and hemagglutination inhibition (HI) activity against influenza viruses with cognate mutations. Results revealed an over 50% reduction in the BLI binding of several mutated HA1 compared to the wild type and a strong correlation between dominant residues identified by the BLI and HI assays. Our study demonstrates a method to systemically analyze antibody immunodominance in the humoral response to influenza vaccines.

## 1. Introduction

Influenza viruses cause major respiratory diseases, with an estimated three to five million people with severe illness and 290,000–650,000 deaths worldwide each year [1], and vaccination is the primary prophylactic countermeasure to limit influenza infection. Protective humoral immune responses induced by vaccines are largely associated with antibodies (Abs) in serum that predominantly target the external surface protein hemagglutinin (HA) [2]. Seasonal influenza vaccines elicit strain-specific Ab responses mainly against the immunodominant globular head domain of HA, or HA1. Abs directed to hypervariable epitopes in or near the receptor binding site (RBS) of HA1 neutralize viruses by blocking virus binding to cell receptors [3]. Chimeric HA-based universal influenza vaccines induce broadly cross-reactive Abs against the stalk domain of HA, or HA2 [4,5]. The binding of these Abs to the conserved HA2 prevents fusion of the viral and endosomal membranes [6], covers the protease cleavage site between HA1 and HA2 and therefore prevents HA maturation [7], or mediates cytotoxicity of virus-infected cells [8,9]. Due to selection pressure elicited by the immune response to vaccination or infection, seasonal influenza viruses constantly accumulate mutations within the antigenic sites of HA to evade human immunity.

Consequently, the composition of influenza vaccines requires frequent updates so that the antigenicity of the recommended vaccine viruses closely matches that of the variant viruses most predominant in the population [10]. Classically defined antigenic sites on the HA1 have been mapped by tracking amino acid changes in antigenic variants arising naturally or by in vitro selection [11,12,13]. These sites cover multiple overlapping epitopes and are highly variable between viral strains. For the influenza A virus (IAV), five sites (Sa, Sb, Ca1, Ca2 and Cb) on the H1 subtype and five (A through E) on the H3 subtype of HA have been mapped [11,12,13]. Accumulation of multiple amino acid substitutions in these sites has been suggested to be associated with substantial antigenic changes [3,12,14]. Further studies on major antigenic changes in influenza A(H3N2) viruses and the immune history of the infected population have found that as little as one amino acid substitution in the HA head can cause antigenic drift of the influenza virus [15,16,17,18,19,20,21]. Therefore, a highly focused Ab response could be elicited by the seasonal influenza vaccine, and the binding of HA to serum Abs might be substantially disrupted by as little as one amino acid substitution.

Identification of dominant binding epitopes of HA for polyclonal serum Abs can help the selection of viral strains for future vaccine formulations [19]. One approach is the hemagglutination inhibition (HI) assay, which is widely used to analyze antigenic variations of circulating viruses [22]. The focused residues, as part of dominant binding epitopes, can be identified by the HI assay using human sera and a panel of mutated viruses, with each variant containing a single amino acid substitution in the HA [15,19]. Mutated viruses with reduced HI titers (≥4-fold) against human sera compared to wild type were selected, and the cognate mutated residues are considered dominant in binding. However, the HI data can be influenced by changes in viral receptor binding avidity [23,24,25,26]. Mutated residues that cause enhanced binding avidities for red blood cells (RBCs) can be falsely defined as dominant. Moreover, the loss of agglutination of RBCs by some viruses also limits the application of this assay [27,28]. The other approach is the microneutralization (MN) assay used to determine titers of neutralizing Abs against HA’s head and stem domains [29]. As with the HI assay, a panel of viruses, each carrying a single amino acid substitution, is tested against human sera by the MN assay. The cognate mutated residues for viruses with reduced MN titers are identified as dominant [16]. Both the HI and MN assays require the generation of recombinant influenza viruses, which is technically demanding and time-consuming. The focused binding residues can also be determined by the generation of escape mutant viruses using monoclonal Abs (mAbs) or serum Abs [30,31]. However, this approach is time-consuming and involves growing viruses that may require special lab facilities and permissions for highly pathogenic strains and potential gain-of-function mutations [32]. Enzyme-linked immunosorbent assay (ELISA) is another approach commonly used to detect the total binding of Abs to antigens (Ags). A panel of mutated recombinant HAs (rHAs), each carrying a single amino acid substitution on the HA, is generated, and their binding to human sera is determined by ELISA. The cognate residues for mutations that cause reduced binding are defined as dominant [33]. The assay is relatively simple and can be used in high-throughput screening. However, non-specific coating of the Ag to a solid phase, such as ELISA plates, could potentially cause false negative/positive results [34,35]. The dominant binding sites can also be analyzed by competitive ELISA, which measures titers of serum to block the binding of mAbs of defined epitopes to the HA [36]. However, the results can be influenced by steric interference between Abs and conformational alterations of HA induced by Ab binding [37,38].

Here, based on a previously described biolayer interferometry (BLI) assay, the flu Ab biosensor assay (f-AbBA) [39], we report on the development of a flexible label-free and cell-free assay, the f-AbBA-2, to determine the dominant binding epitopes of the HA for polyclonal serum Abs. The f-AbBA-2 utilizes an established transient expression system for producing rHA1 in conjunction with a label-free BLI technology [40,41]. By analyzing the binding of human sera to a panel of rHA1 proteins, each carrying one amino acid substitution within or near the classical H1 antigenic sites, the dominant binding epitopes of HA1 for serum Abs against influenza A(H1N1)pdm09 (abbreviated as pH1N1) vaccine have been determined.

## 2. Materials and Methods

### 2.1. Cloning and Expression of rH1, rHA1 and rHA2

Full-length HA ectodomain (residues 1–501, rH1) of pH1N1 virus A/California/07/2009 (CA/07) was expressed and purified as described previously [42]. A codon-optimized cDNA encoding the HA1 domain (residues 31–311) of CA/07 HA was synthesized (GenScript USA Inc., Piscataway, NJ, USA) and sub-cloned into pIEx-4 vector (EMD Millipore, Burlington, MA, USA) using the In-Fusion HD cloning system (Clontech, Mountain View, CA, USA). All subsequent HA1 constructs, each carrying one amino acid substitution within or near the classical H1/H3 antigenic sites or the receptor binding site, were generated from this wild type (WT) pIEx-4-HA1 clone using the QuickChange Lightning Site-Directed Mutagenesis Kit (Stratagene, La Jolla, CA, USA). The point mutations were designed to induce significant changes in the targeted residues’ size and/or charge. The resulting constructs (Table 1) were transiently transfected into Spodoptera frugiperda Sf9 cells (EMD Millipore, MA) using the Cellfectin II transfection reagent (Thermo Fisher Scientific, Waltham, MA, USA). All procedures were performed following protocols provided by the manufacturers. The transfected cells were grown in suspension on an orbital shaker at 27 °C for five days. All rHA1 proteins contained a signal sequence for secretion, a thrombin site at the C-terminus followed by a trimerizing sequence (foldon) from the bacteriophage T4 fibritin for generating functional trimers, and a His-Tag to aid detection. The expression levels of rHA1s secreted in the culture supernatant were quantified by chemiluminescent western blot analysis using Penta-His antibody (Qiagen, Germantown, MD, USA) and a ChemiDoc MP imaging system (Bio-Rad, Hercules, CA, USA) following the manufacturer’s directions. The concentrations of mutant rHA1 determined by densitometric analysis were normalized to the wild-type rHA1, and similar amounts of rHA1s were applied in epitope mapping analysis without further purification. A codon-optimized cDNA encoding the HA2 domain (residues 1–33, 312–386, 420–501) of the mature HA gene of pH1N1 virus A/Michigan/45/2015 (MI/45) with the linkers for the GEN4 construct, as described by Yassine et al. [43] was synthesized (GenScript USA Inc., NJ). The gene was sub-cloned into a pAc-GP67 baculovirus transfer vector. The final construct contained a signal sequence for secretion, a thrombin site at the C-terminus followed by a foldon sequence for generating functional trimers, and a His-Tag. The rHA2 was expressed in Hi5 cells and purified by standard His-Tag purification and subsequent size exclusion chromatography [44]. The HA2 is highly conserved between CA/07 and MI/45, and there are only four substitutions in the HA2 of MI/45 (I321V, E374K, S451N and E499K) compared with CA/07. Based on the crystal structure of pH1N1 HA (PDB ID 3M6S), residues 321, 374 and 451 are buried inside the HA and inaccessible for direct Ab interactions (residue 499 is missing in the structure). The substitutions are not expected to cause major changes in the bindings of rHA2 to human sera.

### 2.2. f-AbBA-2

Dominant binding epitope mapping of rHA1 for serum Abs was performed by f-AbBA-2 using an Octet RED96 system (Sartorius, San Diego, CA, USA) and instructions provided by the manufacturer. Briefly, the transiently expressed rHA1 proteins of similar amount were coupled to anti-penta-His biosensors by incubating the tip of biosensors into the supernatant of recombinant proteins in kinetics buffer with the usage of a sidekick biosensor immobilization station (Sartorius, CA). Sera were diluted in kinetics buffer, and binding was analyzed by BLI on an Octet RED96 system. The assay steps were defined as baseline for 120 s, association for 300 s, and dissociation for 120 s. Data were analyzed using the system software. Binding to the biosensor tip was measured as a wavelength shift (in nanometers), and end-point binding of the association step was used to quantify the binding avidity of serum Abs for recombinant proteins. The results were presented as a percentage of WT rHA1 binding, and ≥50% reduction in binding was used as a cutoff for considering a particular residue as part of a dominant binding epitope. Binding assays of rH1, rHA1, and rHA2 to serum Abs were similarly performed, except that purified recombinant proteins were used.

### 2.3. Serum Samples

The US adults (n = 20, aged 19–44) were immunized with inactivated egg-grown trivalent influenza vaccines (TIVs) in the 2010–2011 influenza season. The TIVs contained H1 component of pH1N1 A/California/07/2009-PR8-like virus. Paired serum samples were collected pre-vaccination and 21–28 days post-vaccination. These anonymized serum samples were provided by a contract organization, and the use of sera was approved by the National Centers for Immunization and Respiratory Diseases (NCIRD), Centers for Disease Control and Prevention (CDC) Human Subject Research Determination Review. Ferret sera against pH1N1 virus A/California/04/2009 (CA/04) were provided by the Virology, Surveillance and Diagnosis Branch at the CDC. All sera were pretreated with receptor-destroying enzyme (RDE, Denka Seiken, Tokyo, Japan) following the manufacturer’s instructions and diluted to working stocks with phosphate-buffered saline (PBS), pH 7.2.

### 2.4. Influenza Viruses

Influenza viruses were propagated in embryonated eggs or Madin-Darby canine kidney (MDCK) cells. Seven pH1N1 viruses were used in this study, including cell-propagated 6B.1 A/Michigan/45/2015 virus (MI/45), egg-propagated 6B.1 A/Michigan/45/2015, which carries egg-adapted mutation Q223R in HA (Q223R), and five previously reported RG-viruses [44,45]. The five RG-viruses were generated using the HA and neuraminidase (NA) genes from CA/08 and 6 internal genes from A/Puerto Rico/8/1934 (PR8); they were the viruses possessing wild-type CA/08 HA (CA/08), HA-K163Q mutation (K163Q), HA-D127T mutation to add an N-glycosylation motif at N125 (125gly), HA-D127N/N129T mutations to add an N-glycosylation motif at D127 (127gly), or HA-K130 deletion (K130∆).

### 2.5. Hemagglutination Inhibition (HI) Assay

For removal of non-specific inhibitors, the sera were treated with RDE (Denka Seiken, Tokyo, Japan) by diluting one part of serum with three parts of RDE (prepared by re-suspending one bottle of RDE with 20 mL of 0.85% NaCl) and the mixtures were incubated at 37 °C for overnight. The RDE enzyme was inactivated by incubation at 56 °C for 30 min, followed by the addition of six parts of PBS for a final dilution of 1/10. For the removal of non-specific agglutinins, the RDE-treated sera were adsorbed with packed turkey red blood cells (TRBC). HI assays were performed in V-bottom 96-well microtiter plates using 4 HA units (HAU) of virus and 0.5% TRBC as described previously [45]. The highest dilutions of serum samples showing complete inhibition of hemagglutination activity were taken as endpoints.

### 2.6. Statistical Analysis

Statistical analyses were performed using GraphPad Prism 9.4.1 (GraphPad Software, La Jolla, CA, USA). The binding of rH1/rHA1/rHA2 and HI titers of pre- and post-vaccination sera were compared using a two-tailed Wilcoxon matched-pairs signed rank test. Correlations between BLI binding and HI titers were computed by a two-tailed Spearman correlation coefficient (r) test. The binding of K163Q rH1/rHA1 was compared to WT by a two-tailed *t*-test. The binding of the mutated rHA1 panel was compared to WT using one-way ANOVA with Dunnett’s test for multiple comparisons. Significance levels are indicated with asterisks: * *p* < 0.05, ** *p* < 0.01, *** *p* < 0.001, and **** *p* < 0.0001.

## 3. Results

### 3.1. Assay Development with Human Sera to A(H1N1)pdm09 Vaccine

We have previously used the f-AbBA to test the total reactivity of rHAs of the full-length ectodomain, consisting of HA1 and HA2 domains, to human sera, and the binding of rHAs to non-HI Abs has been suggested to cause the disparity observed between the detection level and HI titers for a few sera [39]. The HA1 domain is the main target of the strain-specific Ab response elicited by seasonal vaccines, and the HI assay detects neutralizing Abs with binding sites within or near RBS in the HA1. Compared to rHA, a better correlation between the binding of rHA1 to human sera and HI titers has been observed in a multiplex influenza Ab detection assay [46]. We have hypothesized that the binding results would be more consistent with the HI data if only the HA1-bound Abs were detected in the BLI binding assay. Additionally, the exclusion of the HA2 domain from binding would increase the assay sensitivity. Therefore, we chose to use rHA1 in the binding assay to determine the head region dominant binding epitopes of the HA for serum Abs. To understand the reactivity of different HA domains to serum Abs, purified rH1 and rHA1 of pH1N1 virus A/California/07/2009 (CA/07) as well as rHA2 of pH1N1 virus A/Michigan/45/2015 (MI/45) were tested for their binding to 20 paired human sera before and after vaccination with the 2010–2011 inactivated egg-grown trivalent influenza vaccines (TIVs) containing the H1 component of pH1N1 A/California/07/2009-PR8-like virus. The results are presented as the total wavelength shift (in nanometers) at the end-point of the association step in the assay. Compared to the pre-vaccination sera S1, binding of all three proteins to the post-vaccination sera S2 was significantly higher with 5.0, 25.9 and 2.4-fold increased means for rH1, rHA1 and rHA2, respectively (Figure 1A). The HI titers of sera S2 against pH1N1 reverse genetics (RG)-derived virus A/California/08/2009 (CA/08) were also significantly induced with an increased mean of 25.6-fold over that of the sera S1 (Figure 1B; HI titer means are 18 and 455 for the S1 and the S2, respectively). Comparison of the binding activities with the HI titers revealed strong correlations between the binding of rHA1 and the HI titers with correlation coefficient (r) values of 0.6281 and 0.8063 for the S1 and S2 sera, respectively. No significant correlations were found between the binding of rHA2 and the HI titers, while only the binding of rH1 to the S2, but not the S1, showed a significant correlation with the HI titers (Figure 1C). Furthermore, a better correlation was found between the S2 bindings to rH1 and rHA1 when compared with rHA2 (Appendix A).

To further understand if the use of rHA1 would increase the assay sensitivity, a serum sample that has shown lower binding to the K163Q rH1 mutant and reduced HI titer to CA/08 virus carrying K163Q in HA was selected (Figure 2A,B). The K163Q mutation has been known to cause a reduction in HI titers to the pH1N1 virus. It, therefore, is a good target for analyzing the correlation between epitope binding and the function of sera [18]. In our initial attempts to understand whether changes in BLI bindings of rH1 mutants to human sera correlate with altered sera protective function against the virus, the CA/07 rH1 carrying K163Q was used as the representative mutant for the binding test with human sera. The amounts of WT rH1 or rHA1 coupled to the anti-penta-His biosensors were similar to that of the K163Q mutants of rH1 or rHA1, respectively. Therefore, the difference in bindings to the same serum would be attributed to the changed amino acid between the rH1 or rHA1 cohorts. Bindings of CA/07 rH1 or rHA1 containing K163Q substitution to the serum were substantially reduced to 47% or 8% of the WT rH1 or rHA1, respectively, indicating enhanced assay sensitivity for detecting dominant binding epitopes of serum Abs when rHA1 is used (Figure 2A,B). Compared with their rHA1 counterparts, the bindings of rH1 WT or K163Q mutant to the serum were higher and the difference could be caused by extra bindings of rH1 to the serum Abs targeting HA2 (Figure 2A). We generated a panel of CA/07 rHA1 proteins, each carrying one amino acid substitution within or near known antigenic sites (Table 1). Correct folding of the expressed WT and mutant rHA1s was confirmed by their reactivity to ferret sera against the pH1N1 virus (Figure 3A). The updated BLI binding assay, the f-AbBA-2, consisting of a panel of transiently-expressed His-tagged rHA1s, anti-penta-His biosensors and sera from humans or animals, is applied for epitope mapping of serum Abs.

### 3.2. Epitope Mapping of Human Sera

A panel of 35 mutant rHA1s of CA/07 was initially used to map epitopes of serum Abs by the f-AbBA-2. Due to the low reactivity of WT rHA1 to most pre-vaccination sera (Figure 1A), only the post-vaccination sera were tested for dominant binding epitope mapping. To increase throughput, only one dilution of each serum was used for the analysis. The results were presented as a percentage of WT rHA1 binding, and ≥50% reduction in binding was used as a cutoff for considering a particular residue as part of a dominant binding epitope. The 50% cutoff has been used in our epitope mapping of mAbs against H1 or H3 using full-length rHAs [41]. Our initial test of six human serum samples revealed diversified Ab binding patterns between individuals, and amino acids N125, K130, and K163 were the main focused residues (Figure 3A). Further experiments on another eleven samples using a smaller panel of 11 mutant rHA1s of CA/07 confirmed the diversified response with additional dominant residues being located at S121, D127, K142, K160, S162 and R221 (Figure 3B). The low reactivity of WT rHA1 to sera #10, #14 and #17 was found, while binding to the sera was greatly increased when the Q223R mutant was used as Ag (Table 2). Three distinct binding footprints covering sites Sa, Ca2 and RBS were identified for the sera tested by the f-AbBA-2 (Figure 4). We also analyzed the binding of the rHA1 panel to serum Abs against pH1N1 virus infection in ferrets that have shown different Ab responses from humans [17]. Although no mutated residues causing ≥50% reduction in binding were identified for ferret sera against pH1N1 virus, 57% of the WT binding was found to be associated with N156D, which has previously been shown to be part of an epitope for post-infection ferret serum Abs determined by the HI assay [17].

### 3.3. Determination of Dominant Binding Residues Targeted by HI Abs

Strong correlations were found between the binding of rHA1 and corresponding HI titers (Figure 1C), indicating that the dominant binding epitopes of HA1 are the main targets of HI Abs. Therefore, to verify that the dominant residues defined by the f-AbBA-2 are also predominantly bound by HI Abs, the post-vaccination sera were analyzed by the HI assay using a panel of pH1N1 viruses containing one or two amino acid changes in HA (Table 3). Mutated viruses that displayed ≥4-fold reduction or increase in HI titers compared to WT virus were selected, and the cognate mutated HA residues were defined as dominant sites for HI Abs. The focused binding residues for all the sera except for #2 defined by the HI assay, showed good agreement with those by the f-AbBA-2 (Table 4). Interestingly, the HI titers of sera #2, #10 and #17 against the RG virus carrying K130 deletion in HA (K130∆) showed ≥4-fold increase over WT, while a minimal increase in binding of rHA1 K130E mutant to sera #2 was found (Figure 3B).

## 4. Discussion

We have developed a flexible label-free and cell-free assay, the f-AbBA-2, to map dominant binding epitopes of HA for human serum Abs to the pH1N1 vaccine. Results from this BLI-based method generally match that of the HI assay, demonstrating the validity of this approach. Using rHA1 as an antigen increased the assay sensitivity, especially for sera with a high proportion of HA2-bound Abs and allowed the detection of residues that have a minor impact on the binding of epitopes.

pH1N1 vaccine is different from pre-pandemic seasonal A(H1N1) vaccines in that it induces cross-reactive recall Ab response to both the HA1 and HA2 domains in some individuals naïve to pH1N1 virus or vaccine [47]. Our results of rH1/rHA1/rHA2 binding to the sera and HI titers against the pH1N1 virus following the 2010–2011 influenza vaccination confirmed the observation (Figure 1A,B). The pH1N1 virus infection or vaccination histories of the participated individuals before the 2010–2011 influenza season were unknown. Given that the pre-vaccination sera S1 samples were collected over one year post the pH1N1 outbreak (March 2009), some of the vaccinees may have been exposed and seroconverted to the pH1N1 virus before the 2010–2011 influenza season, resulting in the observed bindings of S1 to rH1 and rHA2, as well as rHA1 to a lesser extent. It is also possible that the recombinant proteins could bind to the S1 sera Abs targeting conserved HA epitopes such as the receptor binding sites and stems of HA. These Abs were cross-reactive and could be induced by virus infections or vaccinations before 2009 in some of the vaccinees. The focused Ab responses targeting K130 and K163 of pH1N1 HA have been proposed to be associated with immune imprinting in groups born between 1983–1996 and 1965–1979, respectively [17,18]. Although this study has a small sample size, an age-dependent pattern of immunodominance of the sera can still be discerned from the results; K130-specific response has been found in 4 out of 9 sera (4/9) by the HI assay and 6/9 by the f-AbBA-2 for the group born between 1981–1991, and K163-specific response in 6/10 by both assays for the group born between 1966–1979 (Appendix A). Our findings of Q223R-specific response by the f-AbBA-2 in sera #10, #14 and #17 support the conclusion that egg-grown pH1N1 vaccines containing the avian-adapted Q223R mutation in HA elicit a focused Ab response to the RBS in some vaccinees [20]. The f-AbBA-2 has identified additional residues critical for Ag-Ab interaction, including S121, K142, K160, S162 and R221, that were not detected by the HI assay. Mutations at N125, K142, K160 and R221 have been found in the HA1 of viral isolates or mAb-selected escape mutants of the pH1N1 virus [48,49]. Sera with a focused Ab response to K163 have consistently shown dominant binding at N125, implying that N125 and K163 might belong to the same epitope of one group of Abs binding to antigenic site Sa. Similarly, K142 and R221 might be part of a Ca2 epitope shared by K130 (Table 4 and Figure 4). The current 2022–2023 human influenza vaccine compositions for the Northern Hemispheres have been updated to include pH1N1 viral strains carrying N129D/K130N and S162N/K163Q/S164T mutations of HA that would add an N-linked glycosylation motif at residue 162 [Global Initiative on Sharing Avian Influenza Data (GISAID) isolate accession number EPI_ISL_417210 and EPI_ISL_404460]. These changes could lead to antigenic drift of the pH1N1 virus and altered dominant Ab response to the updated influenza vaccine.

The HI assays have shown ≥4-fold increase in HI titers to the RG-K130∆ virus (for sera #2, #10 and #17) and the egg-grown MI/45 Q223R virus (for sera #2, #5, #11, #13 and #16), while minimal changes in binding levels for the corresponding Ag-Ab interactions have been detected by the f-AbBA-2. The observed disparity between the focused binding sites defined by the f-AbBA-2 and the HI assay remains to be determined. The HI assay detects neutralizing Abs that block the attachment of the influenza virus to receptors on RBCs, while the f-AbBA-2 identifies total HA1-bound Abs that include both neutralizing and non-neutralizing Abs. The results of the HI assay are known to be influenced by alterations in viral receptor binding avidity [23,24,25]. Interestingly, K130N has been suggested to cause decreased receptor binding avidity [17]. It is unknown if K130∆ would similarly affect receptor binding avidity and therefore result in increased HI titers. However, the increased HI titers to the MI/45 Q223R virus could not be attributed to altered receptor binding avidity. Q223R mutation has been shown to cause increased avidity for avian receptors and thus lowered HI titers to the virus [50,51].

One of the major limitations of our method is the lack of full coverage of potential epitopes. For example, of the 20 tested post-vaccination sera against the pH1N1 vaccine, the #12-S2 was the only serum sample that failed to show any significant changes in binding to the rHA1 panel (Figure 3A). This could be due to novel residues predominantly bound by the #12-S2 serum Abs but not covered by the rHA1 panel. It is also possible that some of the mutated residues in the panel were part of the dominant binding epitope for the #12-S2, but the amino acid changes were not significant enough to disrupt the Ag-Ab interaction. The third possibility is that no immunodominant Ab response was induced in the #12-S2, and multiple amino acid substitutions on the HA are required for a substantial reduction in binding to the serum. A zoom-in strategy might be helpful to increase epitope coverage for the assay. Firstly, a panel consisting of five ∆1 rHA1s, each lacking one classically defined antigenic site would be generated as described previously and applied in the f-AbBA-2 to determine the dominant binding antigenic site [52]. Then a “zoom in” panel of rHA1s, each containing one single amino acid substitution focusing on the identified dominant antigenic site, would be produced, and used to analyze dominant binding epitopes. Therefore, the resolution of the assay could be increased through this strategy. Another limitation is that only one concentration of serum samples was used for the assay. The cutoff percentages for the selection of dominant binding epitopes could be affected by the concentrations of sera used in the test. Further analysis using multiple sera dilutions in future experiments is warranted to confirm the dominance of candidate epitopes.

## 5. Conclusions

Our approach could be flexibly applied to conduct dominant binding epitope mapping for serum Abs to other antigens. A panel of rHA1 consisting of 20 to 30 variants can be produced in less than two weeks and the assay can be performed in a high-throughput format using the Octet RED384 system (Sartorius, CA) [41]. Application of this method in influenza surveillance would help improve vaccine virus selection.

## Figures and Tables

**Figure 1 vaccines-11-01307-f001:**
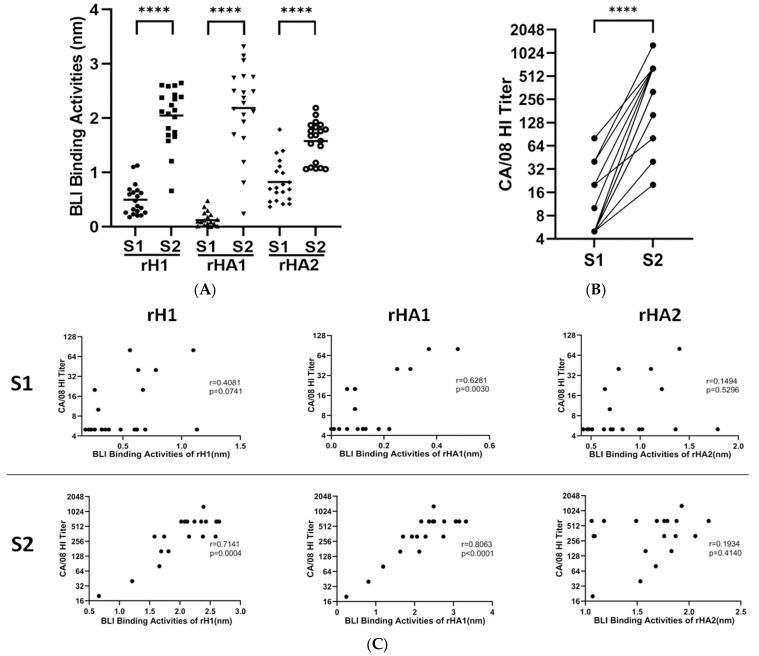
Influenza pH1N1 vaccine-induced Ab response to the HA1 and HA2 domains of HA. (**A**) The binding of rH1/rHA1/rHA2 to the pre-vaccination (S1) and post-vaccination sera (S2) diluted at 1:10 was determined by the f-AbBA. Each symbol represents one measurement of Ag-Ab binding. Horizontal lines show the medians of binding by the group. The results are representative of two independent experiments. (**B**) HI titers of the 20 paired sera to pH1N1 virus CA/08 were determined by the HI assay. Each dot represents one measurement of HI titer for one or multiple serum samples. Results are representative of three independent experiments. (**C**) Correlations of the HI titers to the BLI binding of rH1/rHA1/rHA2 for S1 and S2 were analyzed. Statistical significance was determined using a two-tailed Wilcoxon matched-pairs signed rank test for (**A**,**B**) and a two-tailed Spearman correlation coefficient (r) test for (**C**). Significance levels of **** *p* < 0.0001 are shown in (**A**,**B**), and r plus *p* values are shown in (**C**).

**Figure 2 vaccines-11-01307-f002:**
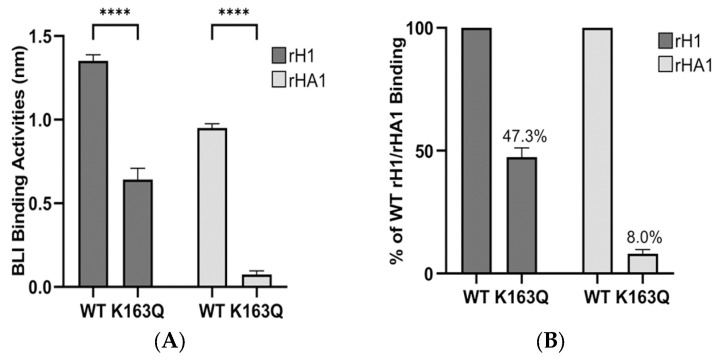
Use of rHA1 as Ag increased the assay sensitivity. (**A**) The binding of rH1/rHA1 of CA/07 WT or K163Q mutant to post-vaccination sera #13 was determined. Each bar represents the median and standard deviation of results from three independent experiments. Statistical significance was determined using a two-tailed *t*-test with significance levels of **** *p* < 0.0001. (**B**) Normalization of binding levels of K163Q mutants to WT of rH1 or rHA1 showed more reduction in the percentage of WT binding using rHA1 K163Q as Ag.

**Figure 3 vaccines-11-01307-f003:**
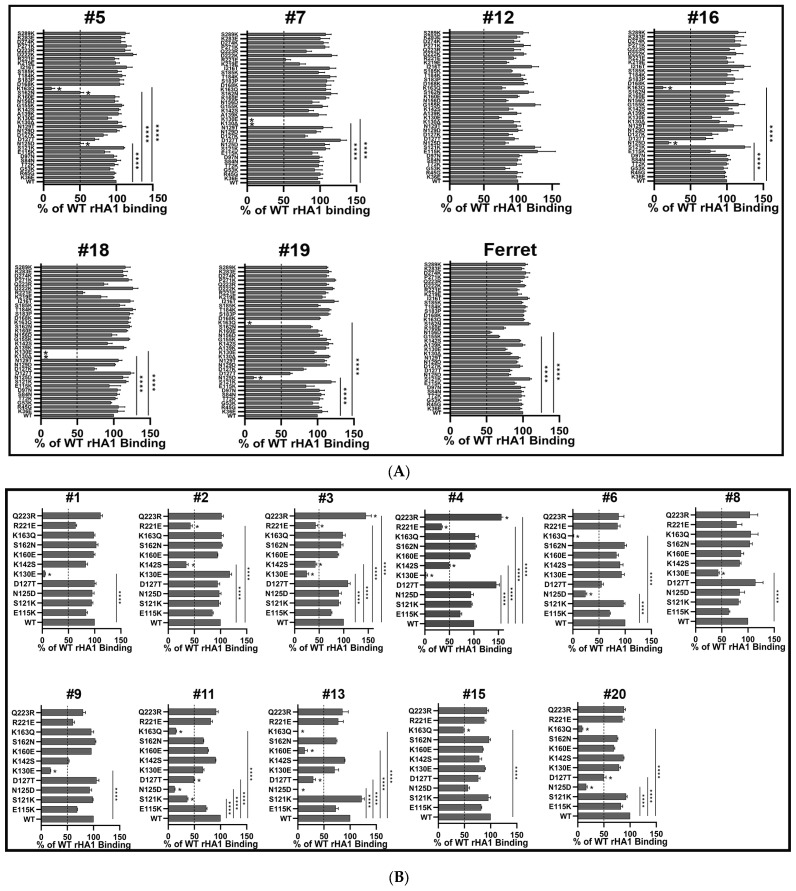
Epitope mapping of post-vaccination sera to pH1N1 vaccine. Dominant binding residues of HA for each indicated serum sample were determined using a panel of 35 (**A**) or 11 (**B**) rHA1 mutants. Each bar represents the median and standard deviation of results from three independent experiments. “*” indicates mutation that causes ≥50% reduction or increase in binding compared to the wild type. The dashed vertical lines indicate the 50% cutoff. Statistical significance was determined using one-way ANOVA with Dunnett’s test for multiple comparisons with significance levels of **** *p* < 0.0001.

**Figure 4 vaccines-11-01307-f004:**
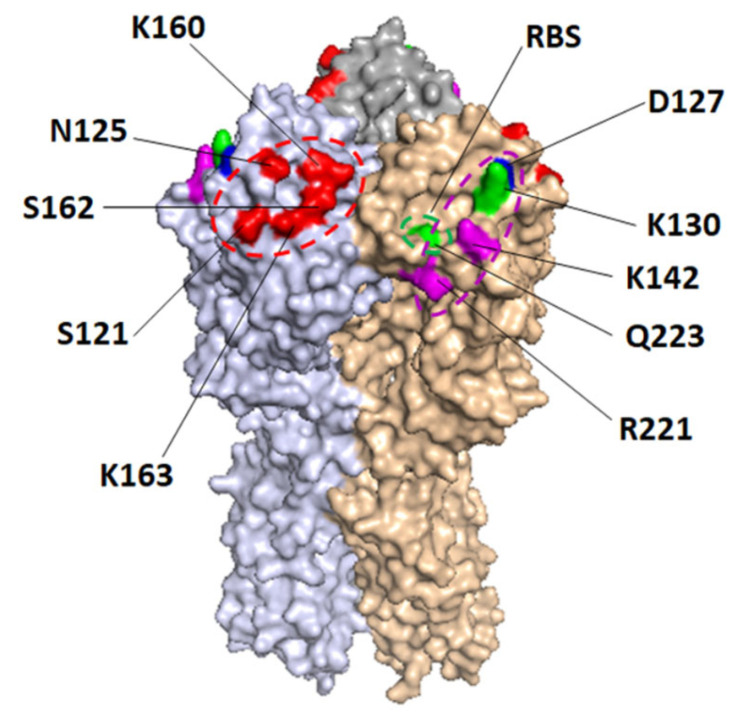
The f-AbBA-2-defined dominant binding epitopes of CA/07 HA in Ab response to pH1N1 vaccine. The sequences and locations of residues as part of dominant binding epitopes are shown on the 3D structure of trimeric pH1N1 HA (PDB ID 3M6S) with different colors indicating corresponding antigenic sites: Sa (red), Ca2 (magenta), and RBS (green). Potential epitopes are indicated by ovals with dashed lines.

**Table 1 vaccines-11-01307-t001:** Mutations of HA in the rHA1 panel of CA/07.

Mutation	H1 Antigenic Site	Equivalent H3 Antigenic Site	Mutation	H1 Antigenic Site	Equivalent H3 Antigenic Site
K36E	- ^a^	-	N156D	Sa	B
R45G	-	-	K160E	Sa	-
G53K	-	E	S162N	Sa	-
T72K	Cb	E	K163Q	Sa	-
S84N	-	E	D168K	Ca1	-
D97N	-	-	S183P	-	B
E115K	-	A	T184K	Sb	B
S121K	-	A	S185K	Sb	B
N125D	Sa	A	I216T	-	D
D127K	-	A	K219E	-	D
D127T ^b^	-	A	R221E	Ca2	D
N129D	-	A	D222K	Ca2	D
N129T	-	A	Q223R	-	D/RBS
K130E	-	A/RBS	P271K	-	C
K130∆	-	A/RBS	D274K	-	C
A139K	Ca2	A	K283E	-	-
K142S	Ca2	A	S289K	-	-
G155K	Sa	B			

^a^ “-“ indicates an undesignated antigenic site. ^b^ D127T would add an N-glycosylation motif at N125.

**Table 2 vaccines-11-01307-t002:** Binding activities and HI titers of post-vaccination sera #10, #14, and #17 against WT and Q223R rHA1 or viruses.

	BLI Binding Activities (nm)		Ratio	HI Titer ^a^		Ratio
Sample ID	CA/07 rHA1	Q223R rHA1	Q223R/WT	MI/45	Q223R	Q223R/WT
#10-S2	0.04 (0.02) ^b^	0.85 (0.03)	21.25	5	160	32
#14-S2	0.22 (0.01)	1.18 (0.04)	5.36	40	640	16
#17-S2	0.38 (0.02)	1.02 (0.04)	2.68	20	1280	64

^a^ Cell-grown virus MI/45 and egg-grown virus MI/45 with Q223R mutation in HA (Q223R) were used in the HI assay. The results are representative of three independent experiments. ^b^ BLI binding activities are presented as mean plus standard deviation (in brackets) of results from three independent experiments.

**Table 3 vaccines-11-01307-t003:** HI titers of post-vaccination sera to pH1N1 viruses.

Sample ID	HI Titer by Virus ^a^						
	CA/08	K163Q	125Gly	127Gly	K130∆	MI/45	Q223R
#1-S2	640	640	640	640	**40** ^b^	1280	1280
#2-S2	320	640	320	160	**1280**	80	**640**
#3-S2	320	320	320	**40**	640	320	**1280**
#4-S2	160	160	320	80	160	160	**640**
#5-S2	640	**80**	320	1280	1280	80	**320**
#6-S2	320	**10**	**40**	320	640	5	10
#7-S2	640	1280	1280	**160**	**10**	1280	1280
#8-S2	160	320	320	80	**10**	320	320
#9-S2	320	320	320	**80**	**5**	160	160
#10-S2	20	20	10	10	**160**	5	**160**
#11-S2	640	**20**	**80**	640	640	20	**80**
#12-S2	640	320	640	640	640	320	320
#13-S2	320	**10**	**20**	320	640	10	**40**
#14-S2	40	40	80	20	80	40	**640**
#15-S2	640	**160**	320	640	640	640	320
#16-S2	640	**20**	**40**	640	640	40	**160**
#17-S2	80	80	80	**20**	**640**	20	**1280**
#18-S2	1280	1280	640	640	**40**	1280	1280
#19-S2	640	**10**	**40**	640	1280	5	5
#20-S2	640	**80**	**160**	640	640	40	40

^a^ Reverse genetics-derived recombinant pH1N1 viruses of wild type CA/08 and mutant viruses with K163Q, 125Gly, 127Gly (Gly: addition of an N-glycosylation motif), or K130 deletion (K130∆) in HA were used in the HI assay. Cell-grown virus MI/45 and egg-grown virus MI/45 with Q223R mutation in HA (Q223R) were used in the HI assay as a separate group. The results are representative of three independent experiments. ^b^ HI titers to mutant viruses with ≥4-fold change compared to wild type are highlighted in bold.

**Table 4 vaccines-11-01307-t004:** Summary of dominant binding residues of HA in Ab response to pH1N1 vaccine determined by the f-AbBA-2 and the HI assay.

Donor Number	Birth Year	Dominant Residues of HA in Bindings by Abs ^a^	
		HI	f-AbBA-2
#1	1991	**K130∆** ^b^	**K130E**
#2	1990	Q223R	K142S, R221E
#3	1989	127Gly, **Q223R**	K130E, K142S, R221E, **Q223R**
$4	1984	**Q223R**	K130E, K142S, R221E, **Q223R**
#5	1983	**K163Q**, Q223R	N125D, S162N, **K163Q**
#6	1981	125Gly, **K163Q**	N125D, **K163Q**
#7	1981	127Gly, **K130∆**	**K130∆/E**
#8	1981	**K130∆**	**K130E**
#9	1981	127Gly, **K130∆**	**K130E**
#10	1980	**Q223R**	**Q223R**
#11	1979	**125Gly**, **K163Q**, Q223R	S121K, N125D, **125Gly**, **K163Q**
#12	1978	N.D. ^c^	N.D.
#13	1976	**125Gly**, **K163Q**, Q223R	N125D, **125Gly**, K160E, **K163Q**
#14	1974	**Q223R**	**Q223R**
#15	1971	**K163Q**	**K163Q**
#16	1970	125Gly, **K163Q**, Q223R	N125D, **K163Q**
#17	1970	127Gly, **Q223R**	**Q223R**
#18	1969	**K130∆**	**K130∆/E**
#19	1969	125Gly, **K163Q**	N125D, **K163Q**
#20	1966	**125Gly**, **K163Q**	N125D, **125Gly**, **K163Q**

^a^ Dominant binding residues of HA identified for post-vaccination serum Abs. ^b^ Dominant binding residues identified by both assays are indicated in bold. ^c^ Not determined.

## Data Availability

All data is freely available upon request to the corresponding authors.

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
