# Peer review of "Use of Biolayer Interferometry to Identify Dominant Binding Epitopes of Influenza Hemagglutinin Protein of A(H1N1)pdm09 in the Antibody Response to 2010–2011 Influenza Seasonal Vaccine"

_vaccines, 2023, doi:10.3390/vaccines11081307_

Round 1

Reviewer 1 Report

The authors of the paper titled "Use of Biolayer Interferometry to Identify Dominant Binding Epitopes of Influenza Hemagglutinin Protein of A(H1N1)pdm09 in the Antibody Response to 2010-2011 Influenza Seasonal Vaccine" discuss a new technique based on biolayer interferometry and binding assays for epitope mapping in influenza surveillance and vaccine virus selection. The paper is well-written and technically correct. However, there are some questions that need to be addressed:

  1. 1- Why did you use rHA2 from an H1/Mich/2015 instead of H1N1/Cal09? It would have been more interesting, especially because the stem HA2 is more conserved than the head HA1.

  2. 2- In Figure 1, why do you observe binding for pre-vaccination samples S1 for rH1 and rHA2? Are these from natural immunity?

  3. 3- It would be nice to include correlation curves in Figure 1 to better illustrate the significance of the positive correlation between S2 and rH1/rHA1.

  4. 4- In Figure 2, is the difference in serum BLI binding activities to rH1 and rHA1 (WT and K163Q) the actual (non-evaluated) binding to rHA2? This result is somewhat confusing and requires further discussion.

  5. 4- Figure 3 should be presented before Table 2.

  6. 5- Table 5 is not explained in the Results section but in the Discussion. It might be better suited as a supplemental figure.

Author Response

Response to Reviewer 1 comments

1- Why did you use rHA2 from an H1/Mich/2015 instead of H1N1/Cal09? It would have been more interesting, especially because the stem HA2 is more conserved than the head HA1.

Author Response: Thanks for the suggestion. The rHA2 of MI/45 was used due to its availability. The HA2 is high conserved between CA/07 and MI/45. Compared with CA/07, three of the four substituted residues in MI/45 HA2 (I321V, E374K and S451N) are buried inside the HA and inaccessible for direct Ab interactions [the fourth substituted residue, E499K is missing in the H1pdm crystal structure (PDB ID 3M6S)]. These substitutions are not expected to cause major changes in the bindings of HA2 to human sera. The relevant information has been updated in Materials and Methods, Lines 131 to 137.

2- In Figure 1, why do you observe binding for pre-vaccination samples S1 for rH1 and rHA2? Are these from natural immunity?

Author Response: Thanks for the comment. The pH1N1 virus infection or vaccination histories of the participated individuals prior to the 2010-2011 influenza season were unknown. Given that the pre-vaccination sera samples S1 were collected over one year post the pH1N1 outbreak (March 2009), it is possible that some of the vaccinees had been exposed and seroconverted to the pH1N1 virus before the 2010-2011 influenza season, resulting in the observed bindings of S1 to rH1 and rHA2, as well as rHA1 to a lesser extent. It is also possible that the recombinant proteins could bind to the S1 sera Abs targeting conserved HA epitopes such as the receptor binding sites and stems of HA. These Abs were cross-reactive and could be induced by virus infections or vaccinations prior to 2009 in some of the vaccinees. Response to these questions has been added to Discussion, Lines 374 to 383.

3- It would be nice to include correlation curves in Figure 1 to better illustrate the significance of the positive correlation between S2 and rH1/rHA1.

Author Response: Thanks for the suggestion. The figure has been added as Figure S1, and at Lines 234-235.

4- In Figure 2, is the difference in serum BLI binding activities to rH1 and rHA1 (WT and K163Q) the actual (non-evaluated) binding to rHA2? This result is somewhat confusing and requires further discussion.

Author Response: Thanks for the comment. The amounts of WT rH1 or rHA1 coupled to the anti-penta-His biosensors were similar to that of the K163Q mutants of rH1 or rHA1, respectively. Therefore, difference in bindings to the same serum would be attributed to the changed amino acid between the rH1 or rHA1 cohorts. Compared with their rHA1 counterparts, the bindings of rH1 WT or K163Q mutant to the serum were higher and the difference could be caused by extra bindings of rH1 to the serum Abs targeting HA2. Further data interpretation has been added to Results, Lines 261 to 264 and Lines 268 to 270.

5- Figure 3 should be presented before Table 2.

Author Response: Thanks for the comment. This has been updated.

6- Table 5 is not explained in the Results section but in the Discussion. It might be better suited as a supplemental figure.

Author Response: Thanks for the comment. This has been edited as Table S1.

Reviewer 2 Report

The manuscript by Zhu Guo et al., describes the development of a cell-free assay, the f-AbBA-2, to determine the dominant binding epitopes of the HA for human polyclonal serum Abs. Human sera bind to a panel of rHA1 proteins, each carrying one amino acid substitution within or near the classical H1 antigenic sites, the dominant binding epitopes of HA1 for serum Abs against influenza A(H1N1)pdm09. This is an interesting finding, the application of this method in influenza surveillance would be very helpful in  improving the selection of the virus to be included in the influenza vaccine.

Author Response

We thank Reviewer 2 for your comments.

Reviewer 3 Report

Authors have developed a label-free and cell-free assay for mapping dominant binding epitopes of influenza HA for human serum antibodies to pH1N1. The BLI binding activity of Abs to rHA1 was strongly correlated with the HI titers, indicating the validity of this method. This paper is well written overall, but should be carefully explained so that it can be understood by readers who are not influenza specialists. There are some concerns that the authors should modify in this manuscript before acceptance.

Line 234 and Figure 2:

Please state clearly the reason why the authors chose the K163Q mutation for the study in Figure 2.

Line 239 and Table 1:

Please state clearly how authors chose these 35 sites and why authors replaced them with those amino acids when preparing the mutants.

Reviewer 4 Report

1.  HI titer is the key data in this study, so the detailed test procedures of the HI assay should be listed in the manuscript, such as about the "remove  non-specific inhibitors",  the source and  working concentration of the RDE, et al. Because the related information coould not be found in the reference 45.

2. The  data basis of the " 50% cutoff"  shoud be illustrated in the manuscript. In additon, whether the "cutoff"  is related to the level of antibody titer to the wild A(H1N1)pdm09 virus?
